# The behavioural and cognitive impacts of digital educational interventions in the emergency department: A systematic review

**Sophie Cleff**[1☺*], **Shubhang Sreeranga**[1☺], **Ibtisam Mahmoud**[1], **Abdullatif Hassan**[1‡],
**Laury Gueyie Noutiamo**[1‡], **Elie Fadel**[1‡], **Jennifer Turnbull**[1,2‡*], **Esli Osmanlliu**[1,2‡]

1 McGill University Health Centre, Montreal, Quebec, Canada, 2 Department of Pediatrics, Division of Emergency Medicine, McGill University Health Centre, Montreal, Quebec, Canada

☺ These authors contributed equally to this work.
‡ AH, LGN and EF also contributed equally to this work. JT and EO also contributed equally to this work.
* sophie.cleff@mail.mcgill.ca (SC); jennifer.turnbull@mcgill.ca (JT)

## Abstract

Ensuring patients and their caregivers understand the health information they receive is an important part of every clinical visit. Digital educational interventions like video discharge instructions, follow-up text messaging, or interactive web-based modules (WBMs) have the potential to improve information retention and influence behaviour. This study aims to systematically evaluate the impact of these interventions on patient and caregiver cognition and behaviour, as well as identify the characteristics of successful interventions and observe how success is measured. In December of 2022, a systematic literature search was conducted in several databases (Cochrane, Embase, MEDLINE (Ovid), Web of Science, ClinicalTrials.gov, and Google Scholar) for randomized controlled trials (RCTs) published between 2012 and 2022. In 2024, an identical search was performed for articled published between 2022 and 2024. Studies testing patient- and caregiver-facing digital educational interventions in the emergency department for behavioural and cognitive outcomes were included. Data from 35 eligible studies encompassing 12,410 participants were analyzed and assessed for bias using the Cochrane RoB2.0 tool. Video was used in 22 studies (63%), making it the most common modality. Seventy-three percent (16/22) of these studies reported statistically significant improvements in their primary outcomes. Text messaging was used in eight studies, with two (25%) reporting significant improvement in their primary outcomes. WBMs and apps were used in seven studies, 71% (5/7) of which reported statistically significant improvements in primary outcomes. Statistically significant improvements in cognitive outcomes were reported in 64% (18/28) of applicable studies, compared with 17% (4/23) for behavioural outcomes. The results suggest that digital educational interventions can positively impact cognitive outcomes in the emergency department. Video, WBM, and app modalities appear particularly effective. However, digital educational interventions may not yet effectively change behaviour. Establishing guidelines for evaluating the quality of digital educational interventions, and the formal adoption of existing reporting guidelines, could improve study quality and consistency in this emerging field. **Registration** The study is registered with PROSPERO ID #CRD42023338771.

**Data availability statement:** All data generated or analysed during this study are included in this published article and its supplementary information files.

**Funding:** The author(s) received no specific funding for this work.

**Competing interests:** The authors have declared that no competing interests exist.

## Author summary

In our study, we explored how emergency department (ED) patients and their caregivers can be educated about important health information through digital tools. We reviewed 35 articles on different methods – such as videos, apps, and text messages – aimed at improving patients' health-related behaviours or their understanding of health issues. Our research is important for several reasons. Firstly, it's crucial that patients and caregivers have a clear understanding of information that could impact their health when they leave the ED – this knowledge supports their future health and well-being. Secondly, digital tools are increasingly being used to convey health information to patients. Rigorous review of these tools supports the creation of effective and accessible interventions for all patients. This not only helps patients and caregivers, but also helps to ensure that the resources to develop such tools are used wisely. Our findings contribute to a broader understanding of how digital educational tools can enhance patient education in the ED.

## Introduction

### Background

Apparent gaps in patient understanding of health information have been studied for decades [1]. Comprehension of standard verbal discharge instructions has been found to be deficient in up to 78% of patients [2], and one study estimated that 60% of Canadian adults have low health literacy [3]. When patients have difficulty understanding complex health information during a clinical encounter, it can lead to improper subsequent care, increased risk of return visits to the hospital, and other poor health outcomes [4–7].

Digital educational interventions show promise for improving health-related outcomes, and may help bridge gaps in understanding for patients that struggle to understand and retain health information when relayed through traditional methods (i.e., verbal discharge instructions or a brochure) [8–10]. Text messaging, video instructions, apps, and Web-Based Modules (WBM) may allow for a more interactive experience with the material, reliable access to important information without a physician present, and a more tailored learning experience than traditional verbal discharge instructions [11]. Apps and WBMs specifically allow for a variety of interactive features such as chatbots, embedded text and videos, behavioural tracking, and questionnaires that help patients engage with the material. Patients can also complete a survey to establish what may help them the most, and the app or WBM can then tailor the information it presents to the needs of the patient. However, the development of new digital educational tools also runs the risk of leaving some low digitally-literate groups even further behind, exacerbating existing disparities [12]. It is thus important to study how these interventions are being implemented and evaluated.

To our knowledge, there has been no comprehensive review of both the behavioral and cognitive impacts of digital educational interventions in emergency medicine. Previous studies have examined solely cognitive impacts [13,14] or impacts in a limited disease area [14], commonly finding evidence that demonstrates some benefits of digital education as compared to usual care. This study will evaluate the cognitive impacts (e.g., information comprehension or confidence in disease management) and behavioural impacts (e.g., changes in health-seeking behaviour or adherence to a medical plan or advice) to determine how digital educational interventions are affecting patients, and guide future development and evaluative efforts in this field.

### Theoretical frameworks

In deciding to focus on behavioural and cognitive impacts, we drew from several theoretical frameworks. The development of our cognitive analysis framework was influenced by the Health Belief Model. This framework proposes that individual perceptions of disease severity, belief in the efficacy of the prescribed action, and confidence in themselves to complete that action can influence health-related behaviours [15]. This directly correlates to several of the subcategories we created for analysis, namely: confidence with disease management, motivation to make behavioural changes, and disease awareness and understanding.

However, there are differences between thought and action; high motivation to change a behaviour does not always result in a behaviour change [16]. Our decision to also investigate changes to health-seeking behaviour is supported by the Normalization Process Theory, which evaluates the success of digital health interventions by emphasizing people's actions over their intentions [17].

Furthermore, some research suggests that improvements to the quality of care are vitally important in determining the success or failure of a digital health intervention. This contributed to our inclusion of clinical outcomes, healthcare facility use, and patient satisfaction as categories for investigation [18].

### Objectives

We developed and achieved two primary objectives. Firstly, to explore how digital educational interventions in the emergency department impact the cognition and behaviour of patients and caregivers. Secondly, to identify the characteristics of successful interventions and determine how success is measured in the studies we examined.

## Materials and methods

We conducted a systematic literature review to fulfill both of our objectives, adhering to the Preferred Reporting Items for Systematic Reviews and Meta-Analysis (PRISMA) guidelines [19]. The protocol for this review can be found in S2 Appendix.

### Inclusion criteria

Randomized Controlled Trials (RCTs) were the sole study design included in the primary analysis. RCTs were included if they tested a digital educational intervention in comparison with usual care, with an analog intervention, or through pre/posttest analysis. Digital interventions are defined in the WHO's third global survey on eHealth as the "use of information and communication technologies in support of health services" [20]. We use this definition with the additional specification that the intervention must be caregiver and/or patient-facing and educational in nature. RCTs were eligible for inclusion if conducted in the emergency department.

### Exclusion criteria

Studies were excluded if they (1) used a study population other than patients or caregivers, (2) used an analog-only intervention, (3) used a digital intervention that was not educational in nature (e.g., telemedicine or telemonitoring), (4) did not aim to investigate behavioural or cognitive impacts of their intervention (e.g., investigated the administrative cost of an intervention), or (5) were conducted without a control or comparator.

## Search strategy

A medical librarian at the McGill University Health Centre searched the following databases: Cochrane, Embase, MEDLINE (Ovid), Web of Science, ClinicalTrials.gov, and Google Scholar. The first search was performed on December 2, 2022, restricted to studied published between December, 2012 and December, 2022. A second identical search was performed on June 28, 2024 to include studies published between December, 2022 and June 2024. The search was restricted to studies available in French or English. We restricted the initial search to the most recent 10 years to keep digital educational interventions relevant to our current context, given the rapid pace of innovation in this area of study. The detailed search strategy is included in S1 Appendix.

## Study selection, data extraction, and data synthesis

The results of the database search were transferred into Rayyan [21] for screening and recording decisions about eligibility. Two blinded student researchers (SC, SS) independently screened all articles for pre-determined eligibility criteria, and conflicts were resolved by two senior researchers (EO, JT). We initially screened articles based on title and abstract. Included articles were then screened based on the full text. Finally, the two student researchers checked the reference lists from all included papers for any additional studies that may have been missed in the initial search. Data extraction was divided between the two student researchers for independent completion. We created an extraction table in Excel to record relevant data on study characteristics and results related to our objectives.

We defined cognitive impacts as relating to changes in patient thought and feeling. This category was subdivided into information comprehension, motivation to make behavioural changes, confidence with disease management, and patient satisfaction. Behavioural outcomes were defined as being related to patient behaviour and were subdivided into health-seeking behaviour (e.g., return visits to the ED) and adherence and concordance to medical plan or advice (e.g., contraception use or medication adherence). We also collected data on clinical outcomes, which was reported in several studies in our review. Results were also subdivided by study population age; adult populations were defined as being 18 years or older. Data were recorded qualitatively in the spreadsheet and later turned to quantitative data to describe the characteristics of studies in terms of cognitive or behavioural outcomes across the intervention modalities of video, text, WBM, or app. Bar graphs were generated from the quantitative analysis to visually represent our pool of studies.

To accurately represent the reported primary outcomes of eligible studies, we categorized studies as either showing (1) significant improvement (statistically significant improvement in a given outcome for the intervention group); (2) non-significant change(conflicting in their findings or reporting no statistically significant difference in the outcome between the intervention and control group); or (3) significant worsening (a statistically significant worsening of a given outcome for the intervention group). Statistical significance was determined by examining the reported data in published articles. The percentages of studies that fell into each of these categories, stratified by modality, cognitive vs behavioural, and adult vs pediatric, were then presented.

## Risk of bias assessment

To assess the risk of bias in our group of studies, one student researcher (SS) independently used the Cochrane RoB2.0 tool for RCTs. The overall risk of bias of each study was calculated based on the Cochrane guidelines and then visually represented using the RoBvis tool [22].

## Results

### Study selection, data extraction, and data synthesis

The initial search in 2022 yielded 6,389 articles. Following duplicate removal, 6,151 articles remained. The second search in 2024 yielded 1,508 articles with no duplicates. The title and abstract screen yielded 53 studies eligible for inclusion form the first search, and 18 articles from the second search. After full text screening, 24 studies were included in the review from the first search, and 8 articles from the second search. We found three additional papers to be included after searching through the references, for a total of 35 included studies (Fig 1) [23–57]. These studies encompass a total of 12,410 participants, with between 33 and 2,521 participants per study. More than half of the studies (20/35) were published in 2020 or after. Most studies (25/35) were published in the USA. About half of included studies (51%, 18/35) had exclusively adult (18+ y/o) patients as their study population. Twenty percent of studies (7/35) targeted both adult and pediatric patients in their study population. The remaining

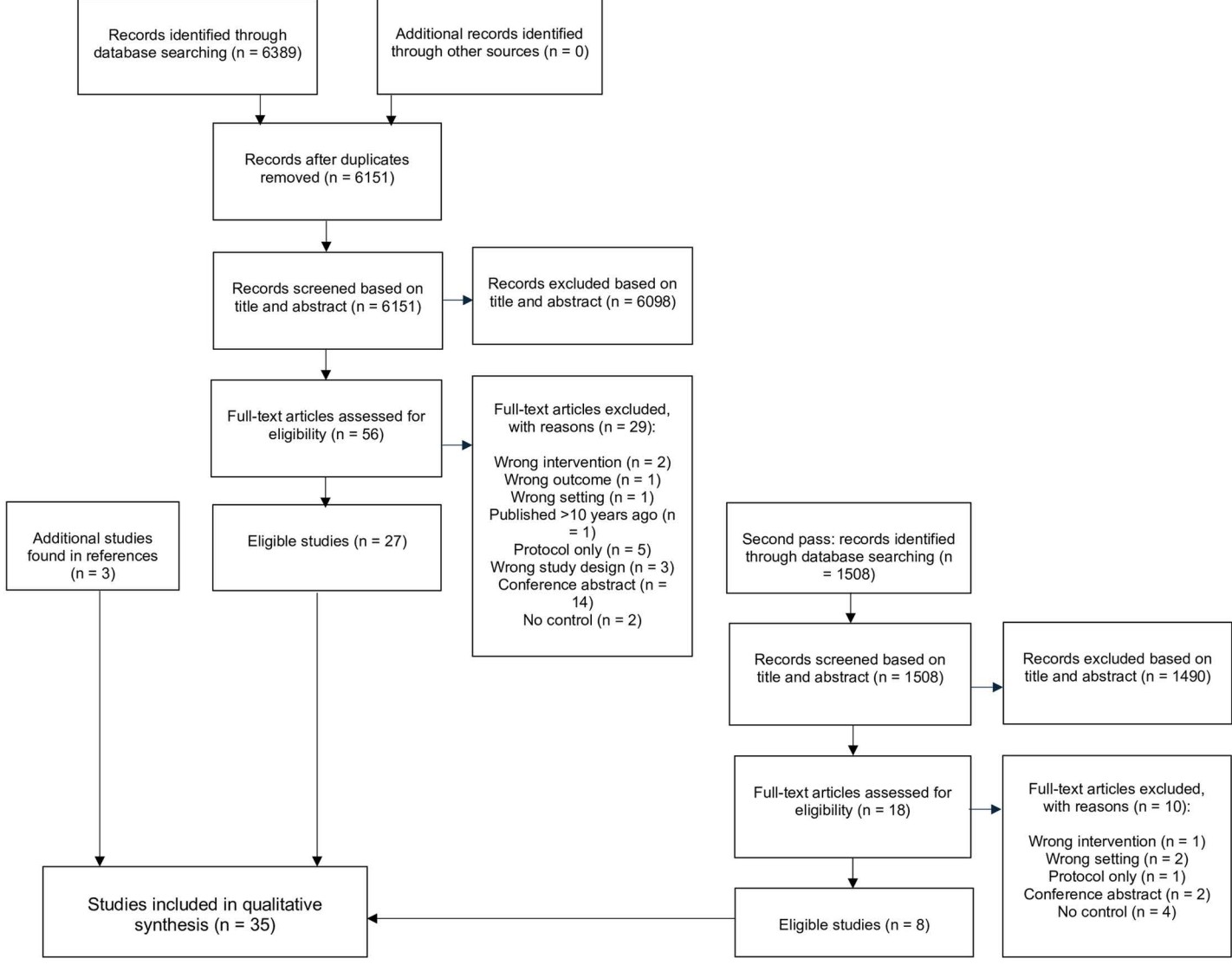

**Fig 1. Preferred Reporting Items for Systematic Reviews and Meta-Analyses (PRISMA) flow diagram for this review.**

29% of studies (10/35) had only pediatric patients. In the studies involving pediatric patients, the child's caregiver received the intervention and was tested for changes in cognitive and behavioral outcomes. The caregiver could also report on clinical and behavioral outcomes for the child. There were no studies testing an intervention exclusively on pediatric patients themselves. The most frequently used digital intervention was video (22/35), followed by text messaging (8/35), apps (3/35), and WBMs (4/35). Out of 35 articles, 23 assessed behavioural impacts, while 28 assessed cognitive impacts (Table 1).

**Table 1. Characteristics of studies in this systematic review.**

| Study characteristics | Number of studies | Study |
|---|---|---|
| *Year of Publication* | | |
| 2013 | 3 | Atzema et al, Bloch et al, Calderon et al |
| 2014 | 1 | Suffoletto et al |
| 2015 | 1 | Chan et al |
| 2016 | 2 | Ismail et al, Olives et al |
| 2017 | 2 | Buis et al, Chernick et al |
| 2018 | 4 | Chakravarthy et al, Golden-Plotnik et al, Ong et al, Platts-Mills et al |
| 2019 | 2 | Belisle et al, Hart et al |
| 2020 | 5 | Lepley et al, Vayngortin et al, Walsh et al, Wilkin et al, Merchant et al |
| 2021 | 5 | Dimeff et al, Hoek et al, Jove-Blanco et al, McElhinny et al, Omaki et al |
| 2022 | 3 | Chernick et al, Meisel et al, Lin et al |
| 2023 | 1 | Rodriguez et al |
| 2024 | 6 | Abar et al, Adler et al, Alqaydi et al, Di Pietro et al, Zhang et al, Rodriguez et al |
| *Country* | | |
| United States | 25 | Bloch et al, Calderon et al, Chan et al, Ismail et al, Olives et al, Buis et al, Chakravarthy et al, Platts-Mills et al, Lepley et al, Vayngortin et al, Walsh et al, Wilkin et al, Dimeff et al, McElhinny et al, Omaki et al, Chernick et al (2022), Chernick et al (2017), Meisel et al, Suffoletto et al, Merchant et al, Abar et al, Adler et al, Zhang et al, Rodriguez et al (2023), Rodriguez et al (2024) |
| Canada | 5 | Atzema et al, Belisle et al, Golden-Plotnik et al, Hart et al, Alqaydi et al |
| Other | 5 | Ong et al, Jove-Blanco et al, Hoek et al, Di Pietro et al, Lin et al |
| *Sample population* | | |
| Adult only (18+ y/o) | 18 | Buis et al, Chakravarthy et al, Chan et al, Dimeff et al, Hoek et al, McElhinny et al, Meisel et al, Omaki et al, Platts-Mills et al, Wilkin et al, Merchant et al, Suffoletto et al, Abar et al, Adler et al, Alqaydi et al, Di Pietro et al, Rodriguez et al (2023), Rodriguez et al (2024) |
| Adult (18+ y/o) and pediatric (<18 y/o) | 7 | Atzema et al, Walsh et al, Vayngortin et al, Calderon et al, Chernick et al (2017), Chernick et al (2022), Olives et al |
| Caregivers of pediatric only (<18 y/o) | 10 | Ismail et al, Jove-Blanco et al, Bloch et al, Belisle et al, Golden-Plotnik et al, Hart et al, Ong et al, Lepley et al, Zhang et al, Lin et al |
| *Outcome Category (studies in bold used this category as their primary outcome)* | | |
| Behavioral | 23 | Ismail et al, Jove-Blanco et al, Vayngortin et al, Belisle et al, Omaki et al, **Suffoletto et al,** Golden-Plotnik et al, **McElhinny et al, Meisel et al,** Platts-Mills et al, **Walsh et al,** Merchant et al, **Buis et al, Chernick et al (2022),** Hoek et al, **Olives et al, Chernick et al (2017),** Lepley et al, **Abar et al, Adler et al,** Zhang et al, **Rodriguez et al (2023), Rodriguez et al (2024)** |
| Cognitive | 28 | **Ismail et al, Jove-Blanco et al, Vayngortin et al, Wilkin et al, Atzema et al, Bloch et al, Calderon et al, Chakravarthy et al, Chan et al,** Belisle et al, **Omaki et al,** Dimeff et al, **Golden-Plotnik et al, Hart et al, Meisel et al, Ong et al, Merchant et al,** Buis et al, Chernick et al (2022), Hoek et al, Olives et al, **Lepley et al, Adler et al, Alqaydi et al, Di Pietro et al, Zhang et al, Rodriguez et al (2023), Lin et al** |
| Clinical | 9 | **Belisle et al,** Omaki et al, Dimeff et al, **Platts-Mills et al,** Buis et al, **Chernick et al (2022), Hoek et al,** Chernick et al (2017), Di Pietro et al |
| *Intervention type (note: Several studies are listed in multiple categories)* | | |
| Video | 22 | Atzema et al, Belisle et al, Bloch et al, Calderon et al, Chakravarthy et al, Chan et al, Golden-Plotnik et al, Hoek et al, Ismail et al, Jove-Blanco et al, McElhinny et al, Meisel et al, Ong et al, Platts-Mills et al, Vayngortin et al, Walsh et al, Wilkin et al, Merchant et al, Di Pietro et al, Rodriguez et al (2023), Rodriguez et al (2024), Lin et al |
| Text messaging | 8 | Buis et al, Chernick et al (2022), Chernick et al (2017), Olives et al, Omaki et al, Suffoletto et al, Abar et al, Adler et al |
| App | 3 | Dimeff et al, Lepley et al, Omaki et al |
| WBM | 4 | Golden-Plotnik et al, Hart et al, Alqaydi et al, Zhang et al |

### Risk of bias assessment

The results of the risk of bias assessment showed that 25 studies (71%) had some concerns of bias; this was most often due to deviations from the intended intervention. The second most frequent reasons for some risk of bias were missing outcome data (often due to loss to follow-up) or bias in the outcome measurement. Eight studies (23%) had a high risk of bias, and only two studies had a low risk of bias (Fig 2).

### Intervention modalities

**Videos.** Video interventions were the most frequently used intervention (22/35) (Table 1). Videos were either animated or made with live actors, usually presenting adaptations of the standard written discharge instructions or depictions of a fictional scenario. Videos were shown to patients while in the ED, and patients were sometimes given a link to the video so it could be watched again later. Of the 22 studies using video interventions, 16 (73%) reported significant improvements in cognitive, behavioural, or clinical primary outcomes (Tables 2–4).

**Text messaging.** Text messaging interventions were initiated in the ED, but often (7/8) [26,30,31,47,48,56,57] extended for days, weeks, or months past the patient's stay. Six of these studies used unidirectional texting [29–31,47,56,57] and two studies used texting with an interactive component [23,45]. The frequency of texting campaigns ranged from daily to once a week. Three studies used strictly educational content in their texts [29,30,47], and five studies used both motivational and educational content [26,31,48,56,57]. Of the eight studies on texting interventions, two studies (25%) showed significant improvement in their primary outcomes; specifically, in disease awareness and confidence with the treatment plan (Tables 2–4).

**Apps and web-based modules.** Apps and WBMs were the least used intervention modality (3/35 and 4/35, respectively). The rate of significant improvement in the primary outcome for this intervention modality was 71% (5/7), specifically with cognitive outcomes such as disease awareness/knowledge and confidence with the treatment plan [33,37,43,47,55]. All seven of these studies examined cognitive outcomes, and six out of seven included information comprehension as an outcome [33,37,43,47,53,55]. The one remaining study only included patient satisfaction as a cognitive outcome, and this study reported significant worsening in the primary outcome (Tables 2–4).

### Pediatric and adult populations

About half of included studies (51%, 18/35) had exclusively adult (18+ y/o) patients as their study population, 20% of studies (7/35) had a mix of adult and pediatric patients in their study population, and 29% (10/35) had only pediatric patients or their caregivers (Table 1). In the studies involving pediatric patients, the child's parent or caregiver received the intervention and was tested for changes in cognitive and behavioral outcomes. The caregiver could also report on clinical and behavioral outcomes for the child. There were no studies testing an intervention exclusively on pediatric (<18 y/o) patients themselves. Studies with adult-only populations reported significant improvement in their primary outcomes for the intervention group in 61% of studies (11/18) [26,27,32,35,41,43,47,50,52,54,55]. In studies with mixed-age patients, significant improvement in the primary outcome was reported in 43% of studies (3/7) [23,25,39]. Finally, studies with pediatric patients and their caregivers showed significant improvement in the primary outcome for 70% of studies (7/10) [24,28,33,36,37,45,51] (Fig 3 and Tables 3-4).

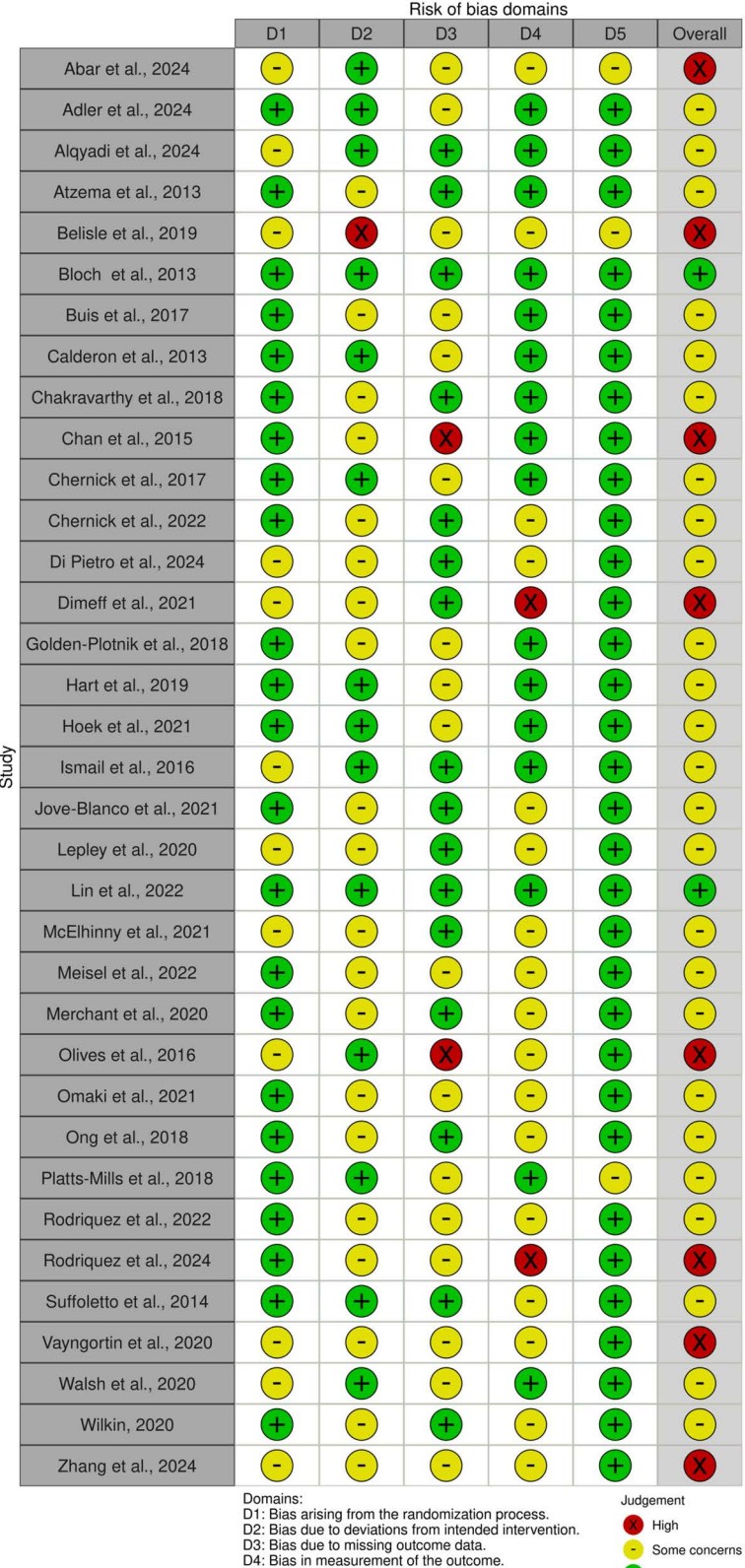

**Fig 2. Results of the Risk of Bias (RoB) assessment, shown through the RobVis tool.**

## Outcomes

**Behavioural impacts.** There were 23 studies that examined behavioural outcomes (Tables 2, 3). These outcomes were measured through self-reporting and were sometimes

**Table 2. A tabular summary visually depicting the outcomes for every study in this review.**

| Author, Year | Cognitive outcomes | Behavioral outcomes | Clinical outcomes |
|---|---|---|---|
| Significant Improvement in Primary Outcome | | | |
| Ismail et al, 2016 | ▲ | ↔ | N/A |
| Jove-Blanco et al, 2016 | ▲ | ↔ | N/A |
| Vayngortin et al, 2020 | ▲ | ↔ | N/A |
| Wilkin et al, 2020 | ▲ | N/A | N/A |
| Atzema et al, 2013 | ▲ | N/A | N/A |
| Bloch et al, 2013 | ▲ | N/A | N/A |
| Calderon et al, 2013 | ▲ | N/A | N/A |
| Chakravarthy et al, 2018 | ▲ | N/A | N/A |
| Chan et al, 2015 | ▲ | N/A | N/A |
| Belisle et al, 2019 | ↔ | ↔ | ▲ |
| Platts-Mills et al, 2018 | N/A | ↔ | ▲ |
| Omaki et al, 2021 | ▲ | ↔ | ↔ |
| Suffoletto et al, 2014 | N/A | ▲ | N/A |
| Dimeff et al, 2021 | ▲ | N/A | ▲ |
| Golden-Plotnik et al, 2018 | ▲ | ↔ | N/A |
| Hart et al, 2019 | ▲ | N/A | N/A |
| Di Pietro et al, 2024 | ▲ | N/A | ▲ |
| Alqaydi et al, 2024 | ▲ | N/A | N/A |
| Rodriguez et al, 2024 | N/A | ▲ | N/A |
| Rodriguez et al, 2023 | ▲ | ▲ | N/A |
| Lin et al, 2022 | ▲ | N/A | N/A |
| Non-significant Change | | | |
| McElhinny et al, 2021 | N/A | ↔ | N/A |
| Meisel et al, 2022 | ↔ | ↔ | N/A |
| Ong et al, 2018 | ↔ | N/A | N/A |
| Walsh et al, 2020 | N/A | ↔ | N/A |
| Merchant et al, 2020 | ↔ | ↔ | N/A |
| Buis et al, 2017 | ↔ | ↔ | ↔ |
| Chernick et al, 2022 | ↔ | ↔ | ↔ |
| Hoek et al, 2021 | ↔ | ↔ | ↔ |
| Olives et al, 2016 | ↔ | ↔ | N/A |
| Adler et al, 2024 | ↔ | ▲ | N/A |
| Abar et al, 2024 | N/A | ↔ | N/A |
| Zhang et al, 2024 | ↔ | ▲ | N/A |
| Significant Worsening in Primary Outcome | | | |
| Chernick et al, 2017 | N/A | ▼ | ↔ |
| Lepley et al, 2020 | ▼ | ↔ | N/A |

▲ Significant Improvement

▼ Significant Worsening

↔ No Change/Conflicting Findings

**Table 3. A tabular summary describing the outcomes of studies examining behavioural outcomes in this review. Bolded text indicates the primary outcome for a given study.**

| Intervention Type | Author, Year | Population and Sample Size | Behavioural Outcome Type | Estimated Impact |
|---|---|---|---|---|
| **Significant Improvement** | | | | |
| Video | Rodriguez et al, 2024 | 767 Noncritically ill, adult ED patients who were not vaccinated for influenza this season | **changes to health-seeking behaviour** | **There was a significant difference in influenza vaccine uptake within 30 days of the intervention between the main intervention group and the control group (41% vs 15%, p<0.0001).** There was no significant difference in uptake between the primary intervention group and the secondary intervention group, with a difference of 8.7 percentage points (95% confidence interval, -0.1 to 17.6 percentage points). |
| Video | Rodriguez et al, 2023 | 496 Noncritically ill adult patients who had not previously received vaccines | **changes to health-seeking behaviour** | **There was a significant increase in the number of intervention group participants who received the COVID-19 vaccine within 30 days of their ED visit, as compared to the control group (44 [20.0%] vs 24 [8.7%]; adjusted difference, 7.9 [95% CI, 1.7-14.1] percentage points)** |
| Text Messaging | Suffoletto et al, 2014 | 765 patients (18 to 25 y/o) reporting hazardous alcohol consumption | **concordance with medical plan or advice** | **Between baseline and three months, there were significant differences in the number of binge drinking days between groups. In the texting and feedback group, the change in number of binge drinking days was -0.51 [95% confidence interval [58] -0.10 to -.95]. In the control group, the change in number of binge drinking days was 0.41 [95% CI -.20 to 1.0]. For the number of drinks per day on drinking days, the change from baseline to follow-up was -0.31 [95% CI -.07 to -.55] for the texting and feedback group, versus 0.39 [95% CI.06 to.72]) for the control group.** The proportion of participants with weekend binge episodes was 30.5% [95% CI 25% to 36%) for the texting and feedback group, as compared with 47.7% [95% CI 40% to 56%] for the texting only group; this was a significant difference. There was also a significant difference between these groups on the number of drinks consumed per weekend; the number was 3.2 [95%CI 2.6 to 3.7] for the texting and feedback group, compared to 4.8 [95% CI 4.0 to 5.6] for the texting alone group. |
| WBM | Zhang et al, 2024 | 211 Parents of PED patients aged 0-12 who own and use cars to transport their children | concordance with medical plan or advice | There was a significant difference in the number of participants who reported modifying their child's car restraint at follow-up, with more intervention participants reported behavioural changes (52.5% vs 31.8%,p=0.003). |
| **Non-significant Change** | | | | |
| Video | Hoek et al, 2021 | 1117 adult (18+y/o) patients with mild traumatic brain injury | changes to health-seeking behavior | The difference in ED visits between groups at 3 months was not significant. The difference in visits, as intervention group minus control group, was −1.0 (95% CI −0.6 to 2.6). |
| Video | Platts-Mills et al, 2018 | 57 adult patients (50+ y/o) with musculoskeletal pain | changes to health-seeking behaviour | 12% of patients in the control arm returned to the ED within one month, compared with 16% of the patients in the video only arm, and 10% of patients in the video and telecare arm. This was evaluated by an EMR check and phone call. |
| Video | Belisle et al, 2019 | 219 parents or primary caretakers of children (6 m/o to 17 y/o) whose chief complaint was otalgia in URT, and the treating physician at least 50% certain of a clinical diagnosis of AOM | changes to health-seeking behaviour | The patients in the control group had slightly more return visits to the ED and absenteeism from school, daycare, or work than the intervention group at follow-up, but the difference was not significant. |
| Video | Jove-Blanco et al, 2016 | 139 caregivers of pediatric patients with acute gastroenteritis (1 m/o to 16 y/o) | changes to health-seeking behaviour | The intervention group had more return visits than the control group, but there was no significant difference in return visits. |
| Video | Ismail et al, 2016 | 63 caregivers (18+ y/o) with a child whose diagnosis included fever or closed head injury | changes to health-seeking behaviour | The intervention group had fewer return visits than the control group, but there was no significant difference in return visits. |
| Video | Vayngortin et al, 2020 | 57 female patients (14-21 y/o), mostly women of color, reporting prior sexual activity. The age distribution was not reported | **concordance with medical plan or advice** | **More patients reported initiating contraception at follow-up than the control group, but the difference was not significant.** |

*(Continued)*

**Table 3.** (Continued)

| Intervention Type | Author, Year | Population and Sample Size | Behavioural Outcome Type | Estimated Impact |
|---|---|---|---|---|
| Video | McElhinny et al, 2021 | 67 adult patients (18+ y/o) with possible opioid poisoning/intoxication | **concordance with medical plan or advice** | **Of the patients who were successfully contacted at follow-up, 33% (13/39) of patients who received video education and 25% (5/20) who received written pamphlet education filled naloxone prescriptions. The p-value of the chi-square for this data was 0.53, with a 95% CI.** |
| Video | Meisel et al, 2022 | 1301 adult patients (18 to 70 y/o) with chief complaint of kidney stones or musculoskeletal back pain | **concordance with medical plan or advice** | **There were no significant differences in use of opioids at 90 days post intervention.** |
| Video *primary outcome not specified | Walsh et al, 2020 | 233 female patients (15+ y/o) who were recent victims of SA and participated in SA medical forensic examination within 7 days of assault. The distribution of age was not reported. | **concordance with medical plan or advice** | **At the first time point, the intervention group smoked significantly fewer cigarettes (6.7) than the control group (11.5) (p=0.029). At the second and third time points, there were no significant difference between groups in the number of cigarettes smoked. There were no significant difference in the number of smoking days across groups at any time point.** |
| Video | Merchant et al, 2020 | 1367 adult (18 to 64 y/o) patients, English or Spanish-speaking | concordance with medical plan or advice | There video intervention group scored slightly higher on the behavioral skills assessments, but there were no significant differences between groups. The largest observed change was for repeat HIV testing after a negative test after recent sex (3.6 (95% CI: −0.4, 7.6)). |
| Text Messaging | Buis et al, 2017 | 123 African American patients (18+ y/o) with hypertension diagnosis | **concordance with medical plan or advice** | **At follow-up, the intervention group showed non-significant mean improvements on the Morisky Medication Adherence Scale as compared to usual care (mean change 0.9, SD 2.0 for the intervention group vs mean change 0.5, SD 1.5 for the control group; p=0.26).** |
| Text Messaging | Chernick et al, 2022 | 146 female patients (14 to 19 y/o) | **concordance with medical plan or advice** | **The absolute risk difference [ARD] between the intervention arm and control arm in terms of initiating contraception was 2.69 [95% CI–12.4 to 17.8]; 2.69% more participants in the intervention arm initiated contraception (i.e., 24.6% of intervention participants and 21.9% of control participants had effective contraception initiation) The ARD in sexual behavior was −3.4 [−18.3 to 11.3], where 3.4% more participants in the control group had sex over the past 3 months. The ARD for follow-up to medical care was −5.1 [−22.3 to 12.1].** |
| Text Messaging | Olives et al, 2016 | 2521 patients with a diagnosis of bacterial, viral, or fungal infection for which an outpatient antibiotic was prescribed. Either adults or <18 y/o if accompanied by a caregiver; number of participants in each age range was not reported | **concordance with medical plan or advice** | **There was no statistically significant difference between the intervention group and control group for 72-hour antibiotic retrieval (Pearson $\chi^2 = 5.112$, P =.078). There was also no significant difference between groups with respect to adherence to the antibiotic course (Person's $\chi^2 = 4.595$, P =.100).** |
| Text Messaging | Adler et al, 2024 | 198 ED patients aged 50-80 y/o/ who have not received and were deemed in need of lung cancer screening | **changes to health-seeking behaviour** | **There was no significant difference between intervention and control groups in rates of lung cancer screening post-intervention (15-22% vs 8-18%, depending on the analysis).** |
| Text Messaging | Abar et al, 2024 | 114 ED patients aged 45-75 y/o who had not been screened for colorectal cancer | **changes to health-seeking behaviour** | **There was a 2-3% increase in the scheduling or completion of screening for colorectal cancer among intervention group participants with follow-up data. Among the whole sample, when assuming that those lost to follow-up were not screened, there was 0-2% absolute difference between groups.** |
| WBM and Video | Golden-Plotnik et al, 2018 | 340 caregivers of children with non-operative fracture | changes to health-seeking behaviour | The percentage of caregivers who reported that they did not miss work days to take care of their child and that their child did not miss school days because of pain was lowest in the video intervention group, but the difference was not significant (p = 0.95 and p = 0.43, respectively) |
| App and Text Messaging*primary outcome not specified | Omaki et al, 2021 | 124 adult patients (18+y/o) with injury or pain-related chief complaint | changes to health-seeking behaviour | There was no significant difference in patients reporting returning to the ED because of pain |

*(Continued)*

**Table 3.** (Continued)

| Intervention Type | Author, Year | Population and Sample Size | Behavioural Outcome Type | Estimated Impact |
|---|---|---|---|---|
| **Significant Worsening** | | | | |
| Text Messaging | Chernick et al, 2017 | 100 female patients (14 to 19 y/o) who were sexually active with males in the past 3 months | **concordance with medical plan or advice,** changes to health-seeking behaviour | **Contraception initiation occurred in 6/50 (12.0%) in the intervention group and 11/49 (22.4%) in the control group.** 16/50 (32.0%) in the intervention group and 15/49 (30.6%) in the control group attended family planning follow-up; 24 in the intervention group (24/50; 48.0%) and 23 in the control group (23/49; 46.9%) received contraception counseling. |
| App | Lepley et al, 2020 | 100 parents or legal guardians of children (<12 y/o) | changes to health-seeking behaviour | Between the app group vs control (14% more visits; IRR, 1.14; 95% confidence interval [CI], 0.56–2.34), and book group vs control (22% fewer visits; IRR, 0.78; 95% CI, 0.34–1.74), there was not a significant difference in the rate of ED visits post-intervention. |

**Table 4. A tabular summary describing the outcomes of studies examining cognitive outcomes in this review. Bolded text indicates the primary outcome for a given study.**

| Intervention Type | Author, Year | Population and Sample Size | Cognitive Outcome Type | Estimated Impact |
|---|---|---|---|---|
| **Significant Improvement** | | | | |
| Video | Ismail et al, 2016 | 63 caregivers (18+ y/o) with a child whose diagnosis included fever or closed head injury | **information comprehension** | **The video intervention group had a significantly higher percentage of correct answers on post-test questionnaires related to discharge instructions and disease understanding (88.89% vs 79.73%, p<0.0001).** |
| Video | Jove-Blanco et al, 2016 | 139 caregivers of pediatric patients with acute gastroenteritis (1 m/o to 16 y/o) | **information comprehension,** patient satisfaction | **The intervention group improvement in knowledge score on the video discharge instructions from pre to posttest was 1.2 (SD 1.11), as compared to 0.94 (SD 0.94) for the control group; this was on a 95% CI and p<0.001.** Patient satisfaction scores were (9.74 SD 0.5) for the intervention group versus 9.64 (SD 0.66) for the control (p = 0.50) |
| Video | Vayn-gortin et al, 2020 | 57 female patients (14-21 y/o), mostly women of color, reporting prior sexual activity. The age distribution was not reported | **motivation to make behavioural changes** | **The mean difference in interest in using an IUD or an implant was significantly higher in the video intervention group; the intervention group's interest score was 1.07 for an IUD, vs 0.297 for the control group (p=0.001, 95% CI). For the implant, the intervention group's interest score was 0.83, vs 0.193 for the control group (p = 0.003, 95% CI)** |
| Video | Wilkin et al, 2020 | 60 military health beneficiaries (18-89 y/o) with upper resp tract infection, pharyngitis or gastroenteritis | **information comprehension** | **There was a significant difference between the video intervention and control group knowledge scores on the post-test questionnaire related to the discharge instructions; the difference was 0.533 (0.14–0.92) in favor of the intervention group, with a 95% CI and p = 0.009.** |
| Video | Atzema et al, 2013 | 133 patients discharged from the ED of any age, with one of 38 diagnoses as determined by an emergency physician. Parents could participate on behalf of their child. The age distribution was not reported. | **information comprehension,** patient satisfaction | **There was a significant difference in mean and median posttest scores in the intervention and control group. The mean score for the video intervention group was 2.5 (s.d 0.8), and for the control group it was 2.1 (s.d. 0.7) (p = 0.002, 95% CI). For the median, the score was 3.0 (IQR 2.0 to 3.0) for the intervention group, while the control group scored 2.5 (IQR 1.5 to 3.0); p=0.001. The patient ratings of the video were not compared between intervention groups.** |
| Video | Bloch et al, 2013 | 436 caregivers of pediatric patients (29 d/o to 18 y/o) diagnosed with wheezing, asthma exacerbation, fever, vomiting, and/or diarrhea | **information comprehension,** patient satisfaction | **There was a significant difference (95% CI) in the mean scores for the video intervention and control groups, both at discharge and 2-5 days post-discharge. At discharge, the mean score for the intervention group was 13.1 (12.5-13.7), whereas the mean score for the control group was 9.1 (8.4-9.7). A few days after discharge, the intervention group scored a mean of 11.5 (10.8-12.3) and the control group scored a mean of 7.5 (6.6-8.3). There was a significant difference in patient satisfaction only a few days after discharge, not at discharge (p<0.05** |

*(Continued)*

**Table 4.** (Continued)

| Interven-tion Type | Author, Year | Population and Sample Size | Cognitive Out-come Type | Estimated Impact |
|---|---|---|---|---|
| Video | Cal-deron et al, 2013 | 203 patients 18-21 y/o (n = 156) or 15-17 y/o (n = 47) who are sexually active and medically stable. | **information comprehension, motivation to make behavioral changes, and confidence in disease management** | **The difference in pre-vs post intervention questionnaires for the video group participants was.98 units more on their intentions to use a condom (95% CI, 0.20–1.77; p =.028), 0.15 units more on condom outcome expectancies 95% CI, 0.07–.23; p =.001), and 0.26 units more on condom use self-efficacy (95% CI,.04–.48; p =.019) compared with the in-person counseling group.** |
| Video | Chakra-varthy et al, 2018 | 55 patients (18+ y/o) with chief com-plaint of pain, unimpaired cognitive function, and receiving outpatient prescription for opioid analgesics | **information comprehension** | **There was a significant difference in the mean post-intervention scores of the intervention group, which scored 21.2/26 (SD = 4.98) on average, and the control group, which scored 16.8/26 (SD = 4.53) on average (p = 0.001).** |
| Video | Chan et al, 2015 | 231 patients (18+ y/o) triaged to urgent care with non-acute illness, primarily African American by convenience sample | **information comprehension** | **The increase in test scores from pre- to post-intervention was highest in the group receiving the combination of all three interventions (video, brochure, and counseling), both immediately after the intervention and at one-month follow-up. The increase in test scores was 72±44.9% for the combination group, 53±48.2% for the video alone group, 45±42.8% for the counseling alone group, and 36±38.1% for the brochure alone group.** |
| Video | Di Pietro et al, 2024 | 114 adult patients (or their care-givers) discharged home with atrial fibrillation or deep vein thrombosis | **information comprehen-sion,** patient satisfaction | **Participants in the intervention group showed significantly higher knowledge of their diagnosis and its potential complications than those in the control group (mean difference, −2.41 out of 18 possi-ble points for the control group; 95% CI, −3.73 to −1.09; p<0.001).** Knowledge of the prescribed therapy did not significantly differ across groups (mean difference, −0.22 out of 6 possible points; 95%CI,−0.84 to 0.39). Patient satisfaction did not significantly differ across groups (mean difference,−0.625 out of 12 possible points; 95%CI−1.82 to 0.57). |
| Video | Lin et al, 2022 | 62 Parents of children in the ED who had been recommended procedural sedation and facial laceration | **information comprehen-sion,** patient satisfaction | **Mean post-intervention knowledge scores were higher in the intervention group 91.67 ± 12.70) than in the conventional group (73.33 ± 19.86), and the difference between pre-intervention and post-intervention scores were significantly greater in the intervention group (coefficient: 18.931, 95% confidence interval: 11.146–26.716).** The intervention group had greater satisfaction than the control group (68.8% of intervention group par-ticipants strongly agreed that the informed consent process was satisfactory, as compared with 30% of the control group participants). |
| Video | Rodri-guez et al, 2023 | 496 Noncritically ill adult patients who had not previously received vaccines | **motivation to make behavioural changes** | **Significantly more intervention participants stated they would get the COVID-19 vaccine in the ED that day if their doctor asked them to (57 [25.8%] vs 33 [12.0%]; adjusted difference, 11.9 [95% CI, 4.5-19.3] percentage points).** |
| Text Messaging | Adler et al, 2024 | 198 ED patients aged 50-80 y/o/ who have not received and were deemed in need of lung cancer screening | confidence with disease management | The intervention group showed significantly improved normative percep-tions of lung cancer screening (LCS). Positive, but statistically insignificant effects on perceived behavioral control over LCS and feelings of competence regarding LCS were also observed in the intervention group. Intervention group participants had higher scores on worries related to LCS. There were no notable differences in feelings of autonomy across groups. |
| App and Text Mes-saging*pri-mary out-come not specified | Omaki et al, 2021 | 124 adult patients (18+y/o) with injury or pain-related chief complaint | information com-prehension, confi-dence with disease management | Knowledge score changes from baseline to discharge were 14.6 (SD = 4.4) to 16.0 (SD = 5.5) in the intervention group, and 15.8 (SD = 4.7) to 16.5 (SD = 5.0) in the control group. These results were not statistically significant between groups. The results for knowledge score changes were also not statistically significant at 6-week follow-up; results were 15.0 (SD = 4.1) to 17.0 (SD = 5.0) in the intervention group, and 15.9 (SD = 5.2) to 17.1 (SD = 4.3) in the control group. **Participants in the inter-vention group reported a significantly greater reduction in decisional conflict about their pain medication preference between baseline and discharge compared to the control group (19.4 point versus 10.4 point drop; F = 3.79; p = 0.05).** There were no statistically significant difference within groups in comfort with the three medication options. 37% of intervention group participants and 50% of control group partic-ipants reported some level of shared decision making with the physician; however, this difference was not statistically significant. |

*(Continued)*

**Table 4.** (Continued)

| Intervention Type | Author, Year | Population and Sample Size | Cognitive Outcome Type | Estimated Impact |
|---|---|---|---|---|
| App *primary outcome not specified | Dimeff et al, 2021 | 33 actively suicidal patients (18+ y/o) seeking ED-based psychiatric crisis services | information comprehension, patient satisfaction, confidence with disease management | A greater proportion of patients in the intervention group reported receiving best practices, as compared to the control group. 14/14 intervention group patients received the crisis stabilization plan, as compared with 2/17 control group patients; 12/14 intervention group patients reported receiving lethal means counseling, as compared with 1/17 control group patients. 13/14 intervention group patients reported receiving 'skills', as compared with 2/17 control group patients; 13/14 intervention group patients reported receiving people with lived experience with suicide education, as compared with 1/12 control group patients (p<0.001). The overall satisfaction rating of intervention group participants was 4.2 (out of 5), whereas the score for the control group was 3.4; the p value was 0.06, indicating no statistical significance. The suicide-related coping score improved by 10 points for the intervention group, whereas it improved by 1.9 for the control group (p<0.001). |
| WBM and Video | Golden-Plotnik et al, 2018 | 340 caregivers of children with non-operative fracture | **information comprehension,** confidence with disease management | **There was a significant difference in knowledge gain score from pre to post intervention among treatment groups. On a 95% CI, the gain score was 2.7(SD = 4) for the video group, 2(SD = 3.1) for the WBM, and 0.4(SD = 2.3) for the control group. No significant difference was found in the confidence to manage pain; 65% in the control group, 62% in the WBM group, and 71% in the video group said they were very confident in managing their child's pain at home (p = 0.4).** |
| WBM | Hart et al, 2019 | 233 primary caregivers of children (1 to 17 y/o) with chief complaint of fever | **information comprehension, patient satisfaction** | **There was a significant difference in knowledge gain score from pre- to post-intervention on a 95% CI for the WBM and ROW groups, as compared with the control (p = 0.001). The difference was 3.4 (SD = 4.2) for the WBM group, 3.5 (SD = 4.1) for the ROW group, and 0.1 (SD = 2.7) for the control group. For patient satisfaction, the WBM group was significantly better than the ROW, and the ROW group was significantly better than the control group (p = 0.001). Out of a max score of 32, the scores were 22.6(SD = 3.2) for the WBM group, 20.7(SD = 4.3) for the ROW group, and 17(SD = 6.2) for the control group.** |
| WBM | Alqaydi et al, 2024 | 79 adult ED (18+) patients with biliary pathology requiring urgent laparoscopic cholecystectomy | **information comprehension, patient satisfaction** | **The intervention group demonstrated significantly higher post-intervention knowledge of the laparoscopic cholecystectomy procedure than the control group (Cohen's d effect size of 0.68 (85.2(10.6)% vs. 78.2(9.9)%; p = 0.004)). Self-reported understanding of the procedure's risks and benefits were significantly higher in the intervention group(Cohen's effect size of 0.76 (68.5(16.4)% vs. 55.1(18.8)%; p = 0.001)). The effects were sustained with a delayed post-intervention assessment(Cohen's effect size of 0.61 (86.5(8.5)% vs. 79.8 (13.1)%; p = 0.024)).** There was no significant difference in patient satisfaction (69.5(6.7)% vs. 67.2(7.7)%; p = 0.149). |
| **Non-significant Change** | | | | |
| Video | Belisle et al, 2019 | 219 parents or primary caretakers of children (6 m/o to 17 y/o) whose chief complaint was otalgia in URT, and the treating physician at least 50% certain of a clinical diagnosis of AOM | information comprehension, patient satisfaction | There were no significant differences in knowledge gain between groups. The level of patient satisfaction was high in both groups. |
| Video | Meisel et al, 2022 | 1301 adult patients (18 to 70 y/o) with chief complaint of kidney stones or musculoskeletal back pain | **information comprehension, patient satisfaction** | **The intervention group demonstrated a similar level of recall of their risk category than the control group (absolute difference = 4.91% of correct recall in favor of the intervention group; 95% CI = -3.0, 12.8).** One day after enrollment, the intervention group reported significantly (p = 0.009) greater satisfaction (rating an average of 7.3 out of 10) than the control group (rating an average 6.6 out of 10), but this difference was not sustained at future follow-up days. |

*(Continued)*

**Table 4.** (Continued)

| Intervention Type | Author, Year | Population and Sample Size | Cognitive Outcome Type | Estimated Impact |
|---|---|---|---|---|
| Video *primary outcome not specified | Ong et al, 2018 | 50 caregivers (18+ y/o) with a child diagnosed in the ED to have typical non-urgent conditions | information comprehension, patient satisfaction | The change in knowledge score from pre-intervention to post-intervention was 1.04 for the control group and 1.56 for the intervention group, with p=0.111 and a 95% CI, indicating statistical insignificance. There was a statistically significant difference in patient satisfaction between the intervention and control group; 80% of the control group participants would recommend the pamphlet to others, and 100% of the intervention group participants would recommend the video to others (p = 0.05) |
| Video | Merchant et al, 2020 | 1367 adult (18 to 64 y/o) patients, English or Spanish-speaking | **information comprehension,** motivation to make behavioral changes, confidence in disease management | **Increases in total mean scores on the knowledge questionnaire from pre to post intervention were 0.43 points (95% CI: 0.07, 0.80) higher for the intervention group than the control group, which was not statistically significant. The change in confidence in recognizing when to be tested was slightly greater in the intervention arm than the control (Δ 0.15; 95% CI: 0.01, 0.28). HIV testing motivation was high before the intervention, and there was not a statistically significant difference in the increase in motivation between groups (Δ 0.05 (−0.03, 0.14)).** |
| Video | Hoek et al, 2021 | 1117 adult patients with mild traumatic brain injury | information comprehension, patient satisfaction | There was no difference between groups in correct recall of the content. In the control group, 31% of patients were highly satisfied about their discharge instructions; in the intervention group, 35.6% were highly satisfied. The statistical significance of this was not noted. |
| Text Messaging | Buis et al, 2017 | 123 African American patients (18+ y/o) with hypertension diagnosis | patient satisfaction, confidence in disease management | There were non-significant improvements in medication adherence self-efficacy (the mean change in the MASES score was 0.8, SD 9.8 for the intervention group and mean change 0.7, SD 7.0 for the control group) (p=0.92). Patient satisfaction was high (94%) in the intervention group, but there was no comparison to the control group. |
| Text Messaging | Chernick et al, 2022 | 146 female patients (14 to 19 y/o) | motivation to make behavioral changes | The absolute risk difference from the intervention group to the control group in intentions to use contraception was 10.0% [95% CI −5.6 to 25.5], where the 10% more of the intervention group seriously considered starting a birth control method in the next 30 days. |
| Text Messaging | Olives et al, 2016 | 2521 patients with a diagnosis of bacterial, viral, or fungal infection for which an outpatient antibiotic was prescribed. Either adults or <18 y/o if accompanied by a caregiver; number of participants in each age range was not reported | patient satisfaction | More participants in the lowest health literacy category preferred voice mailed instructions (14.85% of low health literacy patients preferred voicemail instructions, as compared with 7.41% of high health literacy patients) and more participants in the highest health literacy category preferring written discharge instructions (67.3% of high health literacy patients preferred written instructions, as compared to 51.7% of low health literacy patients) (p = 0.002). |
| WBM | Zhang et al, 2024 | 211 Parents of PED patients aged 0-12 who own and use cars to transport their children | **Information comprehension** | **There was no significant difference between the control group and the intervention group in the increase in knowledge of age-appropriate car restraints at follow-up (74.3% correct in intervention, 61.8% in control, p=0.15).** |
| **Significant Worsening** | | | | |
| App | Lepley et al, 2020 | 100 parents or legal guardians of children (<12 y/o) | **patient satisfaction** | **48.7% of participants receiving the app, 94.6% of participants receiving the book, and 100% of participants in the control group (pamphlet) would recommend the intervention to family or friends (p<0.01)** |

confirmed through electronic medical records [35,50,52,56,57]. Significant improvement in behavioral outcomes were observed in four studies (17%), and non-significant change was observed in 82% of studies (19/23). There were two studies that showed significant worsening of their behavioural outcomes, one of which used a texting intervention and the other of which used an app (Fig 4 and Tables 2–3). Note that there were two studies (Omaki et al. and Golden-Plotnik et al) that were included in multiple intervention modality categories, resulting in 25 data points for Fig 4.

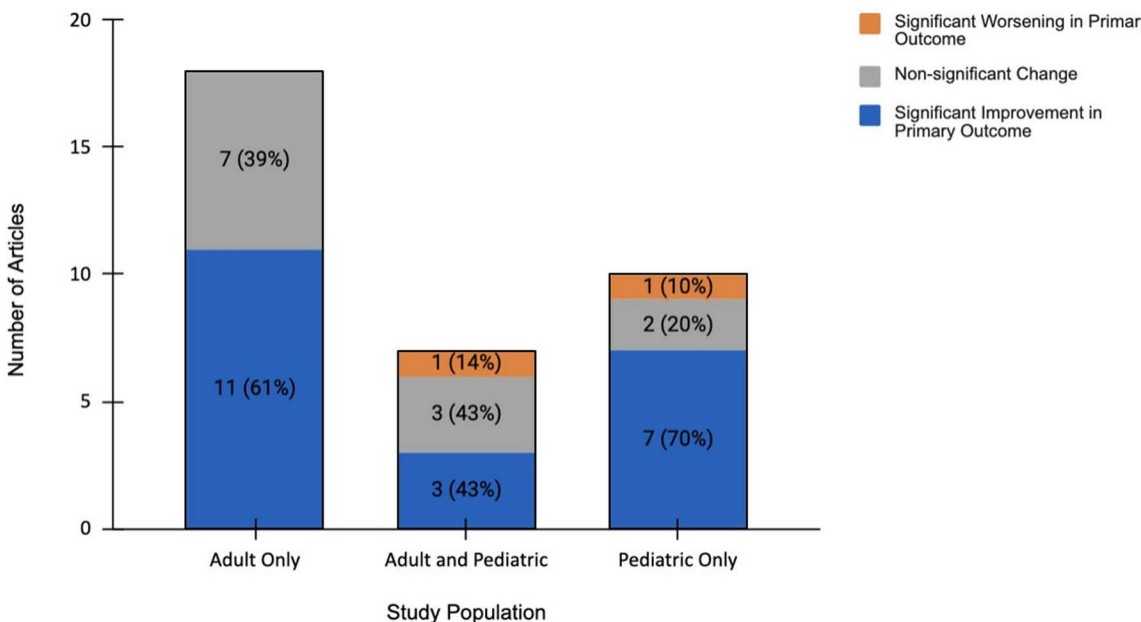

**Fig 3. Primary outcomes of digital educational interventions in adult, age-mixed, and pediatric populations (n = 35).** Percentages indicate the proportion within a given age category.

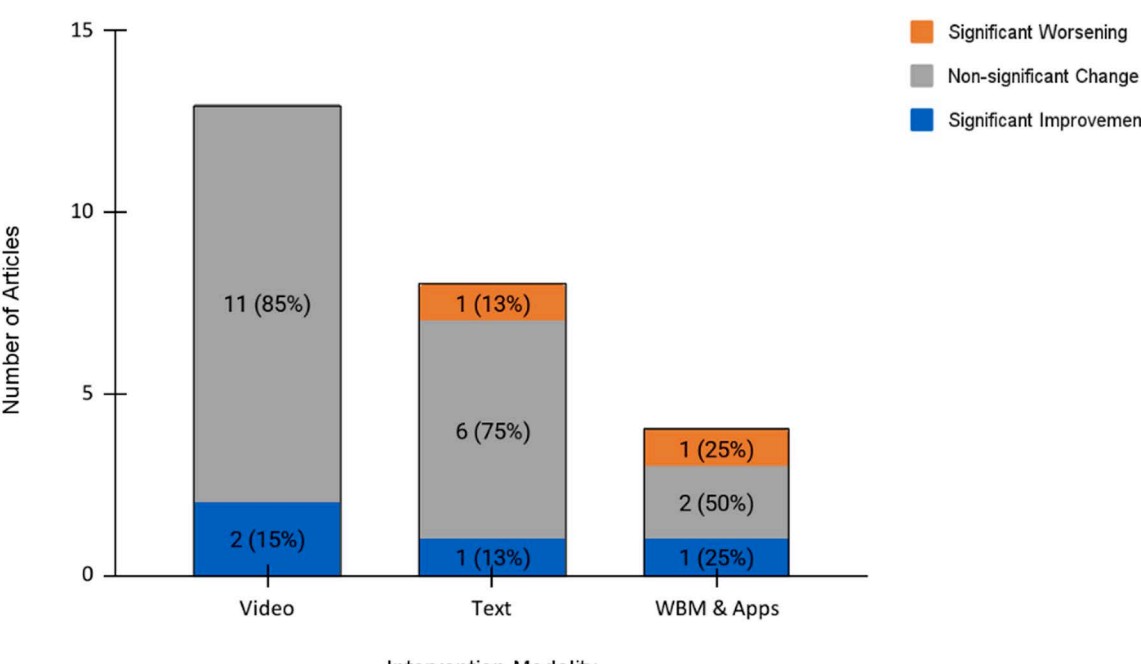

**Fig 4. Stacked bar chart displaying the behavioural outcomes of digital educational interventions in the ED (n = 23).** Percentages indicate the proportion within a given modality.

**Cognitive impacts.** There were 28 studies that examined cognitive outcomes (Fig 5 and Tables 2, 4). The outcomes in this category were mostly measured through knowledge questionnaires and surveys.

Knowledge questionnaires were used in all studies measuring information comprehension (Table 4). They ranged from as few as three questions to as many as 25, varying between multiple choice, true or false, and short answer. Questionnaires were scored by healthcare workers or research assistants. In 12 out of the 21 studies measuring information comprehension, knowledge questionnaires were administered both before and after the intervention [25,27,33,34,36,37,42,45,47,51,53,55]. In the other studies, questionnaires were only administered after the intervention [23,24,28,32,41,43,44,49,54]. In three studies, the questionnaires were administered a third time, up to six weeks post-discharge, to evaluate longer-term knowledge retention [27,47,55]. One of these studies, using a combined app and texting intervention, did not report a significant difference in knowledge retention between the intervention and control group immediately after the intervention or at follow-up [47]. The other two studies, using video [27] and WBM [55] interventions, showed significantly better recall in the intervention group both immediately after discharge and at follow-up.

To evaluate cognitive impacts other than disease awareness (i.e., patient confidence in disease management, motivation to make behavioural changes, and satisfaction), 5-point Likert scales were most often used. Patient-Reported Experience Measures (PREMs) like the Emergency Room Patient Satisfaction Survey were also used to evaluate patient satisfaction [43].

Sixty-four percent of studies (18/28) demonstrated significant improvements in their cognitive outcomes, and in 32% of studies (9/28) there was no change or conflicting findings for the cognitive outcomes between the intervention and control groups. There was one study that demonstrated significant worsening in its cognitive outcome, which used an app intervention (Tables 2, 4). As with the behavioral outcomes, there were two studies (Omaki et al.

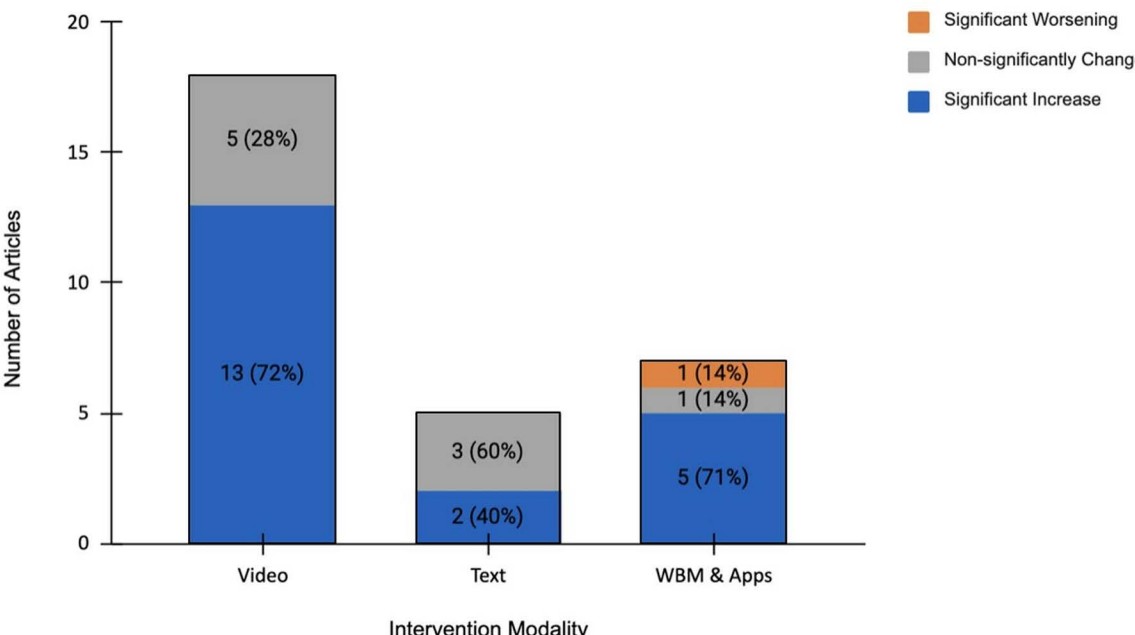

**Fig 5. Stacked bar chart displaying the cognitive outcomes of digital educational interventions in the ED (n = 28).** Percentages indicate the proportion within a given modality.

and Golden-Plotnik et al) that were included in multiple intervention modality categories, resulting in 30 data points for Fig 5.

**Clinical outcomes.** When reporting on clinical outcomes, researchers occasionally (2/8) used Patient-Reported Outcome Measures (PROMs) [43,44]. Examples include the Hospital Anxiety and Depression Scale [44] and the Safety and Imminent Distress Questionnaire [43], but no single PROM was commonly used among multiple studies. In other studies, physicians evaluated physiological aspects of clinical outcomes, for example by administering a test for HIV [25] or a blood pressure test [30]. Forty-four percent of studies examining clinical outcomes (4/9) showed significantly improvement in their primary outcomes. The rest of the studies reported no significant change in their clinical outcomes (Table 2).

## Discussion

### Principal results

In this systematic review, most health-related digital education interventions in the emergency department lead to positive outcomes. The evidence base was stronger for cognitive outcomes (e.g., information comprehension) than for behavioural outcomes (e.g., hospital return visits). Videos were the most frequently used intervention, followed by text-messages and WBMs. The heterogeneous evaluation frameworks used across studies limit the comparison of these tools. A few studies included clinical outcomes, and a minority described any PROMs, which are essential for value-based, patient-partnered digital innovation.

Our primary objective was to explore how digital educational interventions in the ED impact the cognition and behaviour of caregivers and patients. Many of the interventions in these studies focused on educating patients and caregivers; it follows that the researchers would investigate the comprehension of the information given, leading to a strong evidence base for cognitive outcomes. We observed that most interventions investigating cognitive outcomes led to significant improvement (reporting statistically significant improvements for the intervention groups as compared to the control) in their primary outcomes. Many other studies reports non-significant change or conflicting findings in their primary outcomes. There was a notable trend of interventions that had some observable positive effect but did not achieve statistical significance. While this lack of significance in some results could indicate that there is in fact no effect for a given intervention, real effects may have been missed due to lack of power and high dropout rates observed in many of the included studies. Overall, when utilizing digital educational interventions, the likelihood for improved cognitive outcomes appears high.

Interestingly, our analysis demonstrated a difference in efficacy of digital educational interventions for cognitive and behavioural outcomes: interventions appear less effective in influencing behavioural outcomes than cognitive outcomes. Most studies researching behavioural changes reported non-significant change in their primary outcomes, compared with significant improvement in most studies examining cognitive outcomes. These results may be due to complexities behind health-seeking behaviour that go beyond knowledge and understanding. For example, even if a patient understands the health information relayed to them during a visit, their actual health-seeking behaviour may be influenced by practical barriers like access to primary care [59].

In terms of age stratification, statistically significant improvements in both behavioral and cognitive outcomes were achieved in most studies with a solely adult study population, and even more frequently in studies using pediatric populations (<18 y/o) and their caregivers. In studies involving pediatric patients, parents were almost always the ones

who received the intervention [23,24,26,28,33,36–38,45,51]. Thus, the success in studies with pediatric populations should be compared with research involving parents or caregivers of pediatric patients, not research testing interventions on pediatric patients themselves. Previous reviews that have found digital health interventions to be effective in improving parents' health literacy and changing their behaviour [60,61], although consensus has not been reached [62]. There are several potential explanations for why parents and caregivers of pediatric patients showed greater improvement in primary outcomes than adult populations participating on their own behalf. Although the exact age distribution was not reported in most studies, it's possible that the average age of parents taking part in a study on behalf of their children may have been lower than the average age of patients in adult-only studies where participants could be in their 70s and 80s [41,56,57]. If this were the case, the success of interventions in studies using parents and caregivers of children might be explained by higher digital literacy and lower digital exclusion as compared to older populations [63,64]. Furthermore, parents may be motivated to engage in the intervention for the sake of their child's health more than adults are for their own sake [65]. In some of the studies of adult-only patients, a caregiver could answer on behalf of the adult patients, but this was rare [23,54]. However, in the broader literature, there is evidence that educational interventions aimed at caregivers of adult and elderly patients could be effective as well [66,67].

As for interventions for age-mixed populations, significant improvement was reported in less than half of papers included in our review. These age-mixed studies were most often targeting teenagers and young adults with interventions related to sexual and reproductive health. Other research in this field has drawn diverse conclusions. Previous analysis in digital sexual health interventions for young adults has found 75% of studies to be effective in changing behaviour and cognition [68] -- this discrepancy could suggest that the emergency department is not the optimal setting to deliver such interventions. Yet, other research has found that digital health interventions are not effective in influencing the behaviour of adolescents and has called for greater participation of children and adolescents in co-design of future interventions [69]. Interventions designed (or co-designed) specifically for children could be an interesting avenue of future research, especially given that none of the studies in this review directed their interventions solely at a pediatric population.

Our second objective was to identify the characteristics of successful interventions and determine how success is measured in the studies we examined. Among intervention types, we observe that video interventions, closely followed by WBM and app interventions, had the highest rates of significant improvement, with greater improvement in cognitive outcomes than behavioural outcomes. Text message interventions also led to improvement in cognitive outcomes in most studies. This corroborates previous reviews which have found web-[70], video-[71], and text message-[72] based educational interventions to be effective at positively influencing cognitive outcomes. There are several potential explanations as to why videos are the most effective modality. Videos can provide dynamic visualization of complex concepts, while allowing the patient to pause, go back, and generally engage with the material at their own pace. Videos also provide a standardized learning environment where the information can be relayed accurately every time, reducing opportunities for human error in demonstration [73]. Furthermore, the Cognitive Theory of Multimedia Learning suggests that media combining visual and auditory modalities can improve understanding, in comparison with words only [74].

Regarding outcome measurement, the most common metric for success was an improved score on a knowledge questionnaire after receipt of the intervention. Patient satisfaction was also a common measure, although rarely the primary outcome.

### Future directions

**Standardized reporting and evaluation.** There are several opportunities for improvement that we identified for future development, implementation, and evaluation of digital educational interventions in the emergency department. There is an overarching lack of standardization in analysis of intervention quality and in reporting of results. A standardized, widely adopted evaluation framework for such interventions would facilitate the assessment of their quality, a factor which can greatly impact study outcomes, and which is currently difficult to gather data on. Some studies reference known issues with their interventions, such as use of jargon, but many studies make no reference to the perceived quality of their intervention. Several sets of reporting guidelines for digital health interventions exist (e.g., mERA [75], iCHECK-DH [76], and STEDI [77]) to supplement the CONSORT guidelines for RCTs. More widespread adoption of these guidelines would likely increase study quality and integrity, as well as make future studies more easily comparable.

**Intervention accessibility.** It is also worth noting that many of these studies excluded participants with limited digital literacy or lack of phone or internet access, due to the fact that many of these interventions relied on the use of phones or the internet. It serves as a reminder of the risk of exacerbating disparities in health access and outcomes if the deployed technologies are not accessible and inclusive by design, an effect which is often pronounced for people who are underserved, people with disabilities, or those with lower incomes [78]. One strategy to tackle these disparities is to encourage the participation of people who have low technological literacy in studies, giving them controlled opportunities to engage with digital interventions [79]. Increased inclusion of diverse populations in future research would also aid in the generalizability of results and in addressing issues with accessibility early in the design process.

Generally, co-design with patients is an important element to consider for future research into digital educational interventions. Meaningful patient engagement adds value to every step of the research process, from formulating a research question, to outlining a methodology, designing an intervention, conducting an analysis, co-interpreting results, and mobilizing new knowledge [80,81]. Many of the studies in this review discussed having their interventions reviewed by physicians or experts for accuracy, but not by patients for clarity. Retroactive evaluation by patients via satisfaction measures is useful, but future digital educational interventions should engage patients with lived experiences in the ED in their co-design and evaluation [81,82]. These interventions are ultimately for the benefit of patients; they should be partners in the research efforts that create them and measure their impact.

## Limitations

The results of this systematic review must be interpreted in the context of some limitations. Firstly, due to the heterogeneity of articles in our sample, we were unable to perform a meta-analysis. We worked to mitigate this through descriptive quantitative analyses and our thorough qualitative discussion. There was an overall lack of consistency in study methods and reporting of outcomes, which limits cohesive comparison of all articles, but this in itself is a notable finding for future researchers to build on.

## Conclusions

Digital educational interventions have demonstrated the ability to positively impact cognitive outcomes in the emergency department, from information comprehension to patient satisfaction. Videos, WBM, and app interventions seem particularly impactful for caregivers of

pediatric populations. However, digital educational interventions may not yet be effective at changing behaviour.

These interventions are important discharge tools. Increasing patients' understanding of their condition and the care they should take after leaving the ED is important to reduce future injury, reduce return visits to the ED, and improve health outcomes [4,5,6,7]. Future research should work towards developing and enforcing guidelines for the conduct and reporting of digital educational interventions to standardize this quickly growing area of study. Ultimately, these tools should be used to empower patient decision-making and improve health-related outcomes.

## Supporting information

**S1 Appendix.  Search strategies for this systematic review.**
(DOCX)

**S2 Appendix.  Original Protocol for this systematic review.**
(DOCX)

**S1 Checklist.  PRISMA checklist for this systematic review.**
(DOCX)

## Author contributions

**Conceptualization:** Sophie Cleff, Shubhang Sreeranga, Jennifer Turnbull, Esli Osmanlliu.

**Data curation:** Abdullatif Hassan, Elie Fadel, Laury Gueyie.

**Formal analysis:** Shubhang Sreeranga.

**Investigation:** Sophie Cleff, Shubhang Sreeranga, Ibtisam Mahmoud, Jennifer Turnbull, Esli Osmanlliu.

**Visualization:** Shubhang Sreeranga.

**Writing – original draft:** Sophie Cleff.

**Writing – review & editing:** Sophie Cleff, Jennifer Turnbull, Esli Osmanlliu.

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
