## [Decision Letter · Decision Letter 0]

20 Jun 2024

PDIG-D-24-00184

The behavioural and cognitive impacts of digital educational interventions in the emergency department: a systematic review

PLOS Digital Health

Dear Dr. Turnbull,

Thank you for submitting your manuscript to PLOS Digital Health. After careful consideration, we feel that it has merit but does not fully meet PLOS Digital Health's publication criteria as it currently stands. Therefore, we invite you to submit a revised version of the manuscript that addresses the points raised during the review process.

Please submit your revised manuscript within 60 days Aug 19 2024 11:59PM. If you will need more time than this to complete your revisions, please reply to this message or contact the journal office at digitalhealth@plos.org. Please include the following items when submitting your revised manuscript:

We look forward to receiving your revised manuscript.

Kind regards,

Jennifer N Avari Silva, MD

Section Editor

PLOS Digital Health

Journal Requirements:

Additional Editor Comments (if provided):

Please see above comments from R1.

Reviewers' comments:

Reviewer's Responses to Questions

**Comments to the Author**

1. Does this manuscript meet PLOS Digital Health’s publication criteria?

Reviewer #1: Yes

2. Has the statistical analysis been performed appropriately and rigorously?

Reviewer #1: N/A

3. Have the authors made all data underlying the findings in their manuscript fully available (please refer to the Data Availability Statement at the start of the manuscript PDF file)?

Reviewer #1: Yes

4. Is the manuscript presented in an intelligible fashion and written in standard English?

PLOS Digital Health does not copyedit accepted manuscripts, so the language in submitted articles must be clear, correct, and unambiguous. Any typographical or grammatical errors should be corrected at revision, so please note any specific errors here.

Reviewer #1: Yes

Reviewer #1: Thank you for giving me the opportunity to review this paper. Overall, digital interventions are a hot topic and appreciate more studies that are looking at digital interventions in discrete professions/ areas. I will note that "digital interventions" is a very broad topic, but can appreciate the difficulty or lack of evidence to focus only on one modality at this point in time. My major concerns are: 1) the search is outdated, the last time a search was run based on the manuscript was Dec 2022 - this paper cannot be published until a new search is conducted; 2) please change the language around "non-significant improvement" - if there is no statistical significance, there is NO significance, there is no improvement - writing "non-significant improvements" is misleading to readers, especially for those who are not familiar to research, or authors can use "non-significant changes", but definitely not "non-significant success/ unsuccess"

In the introduction:

-authors mentioned in line 70 that "discharge instructions has been found to be limited to some degree 78% of the time.." what kind of limitations are we talking about? Is it staffing limits? Resources? Not quite clear because the latter half of the sentence is talking about patients' literacy;

-authors mentioned that "patient comprehension of health information is low during a clinical encounter" what are you referring to? Patient's lack of understanding because of their low literacy? Being acutely ill?

-the beginning of paragraph 2 details how digital educational interventions can help bridge gaps... etc. but how can digital educational interventions help bridge gaps if patients have inherently low literacy level? If a patient doesn't understand English well, an English video likely isn't going to address this limitation - how can a different modality help? Also, this sentence needs references, can't make claims about how digital interventions can help and provide no citations;

Methods: Greatest concern is this is supposed a systematic review and the latest search was December 2022 - there must be an updated search done within this year before this paper can be published;

-Please provide full details of the library, i.e., Health sciences librarian of Faculty XXX of XX University;

Exclusion criteria:

-mentioned "used physicians as study population" - not sure I understand this exclusion criteria, thought this study was focused on patient/ caregivers, so why would physicians be the only discipline excluded? Or did authors mean that anyone else BESIDES from patients/ caregivers, are excluded?

-#3 digital interventions that did not include education - so are we excluded apps that were meant for physical activity tracking for example? Like would something like Fitbit + app be excluded? please be clear;

Results:

-Table 2, what is value? Is this supposed to be study #? i.e., n=?

-Table 4, like the illustrations in this table, but do not like the 5 different categories used - please change it to successful, unsuccessful or no changes - to my point above that if there is no statistical significance, it is not significant! It's confusing for readers, so stick to the 3 categories, the discussion is where you can talk about how there are studies that show positive/ negative trends that did not reach statistical significance, this would strengthen your discussion points;

-Tables 5/6 - this goes back to your categories, keep it successful, unsuccessful, no change - the 5 categories you have now is extremely difficult to follow (confusing) - it is also not clear to me why certain rows / columns are bolded and others are not;

Discussion:

-to reflect the points above, please change the language of "non-significantly successful" it is "not significant, but there may be potential due to the noted trend" - agree that these may be present due to the lack of high n in these studies (lines 348-350);

-since you added data for pediatrics and adults, there needs to be some discussion about this - were there differences in the # of statistical significantly successful in adults vs children studies? How about the ones that had both populations? there are also age-related factors that may explain the success/ unsuccessful rates, which the authors did not discuss at all. One of the key barrier/ limitation we talk about in digital technology for patients is the "digital divide" - will it actually help those who are older and may not be interested? also what are the age range in these studies that had children?

**Do you want your identity to be public for this peer review?** For information about this choice, including consent withdrawal, please see our Privacy Policy

Reviewer #1: No

---

## [Decision Letter · Decision Letter 1]

30 Dec 2024

Response to Reviewers
Revised Manuscript with Track Changes
Manuscript
**Journal Requirements:**

1. As required by our policy on Data Availability, please ensure your manuscript or supplementary information includes the following: 

**Additional Editor Comments (if provided):**
**Reviewers' Comments:**

**Comments to the Author**

Reviewer #1: (No Response)

publication criteria?

Reviewer #1: Partly

3. Has the statistical analysis been performed appropriately and rigorously?

Reviewer #1: N/A

4. Have the authors made all data underlying the findings in their manuscript fully available (please refer to the Data Availability Statement at the start of the manuscript PDF file)?

Reviewer #1: Yes

5. Is the manuscript presented in an intelligible fashion and written in standard English?

Reviewer #1: Yes

Reviewer #1: Thanks for addressing most of the previous queries. There are a few more things to consider/ address in the discussion section:

1. I will note that there is a paragraph in the discussion that seems underdeveloped. This is the paragraph starting with: "Interestingly, digital educational interventions seem less effective in terms of behavioural outcomes..."

-Is this supposed to say that "educational interventions are not effective for changing behavioural outcomes"?

-I find the message in this paragraph not clear - are you trying to say that health seeking behaviours are not the same as literacy comprehension?

2. The paragraph following that discusses age-differences across studies - there was a claim about assuming parents are the ones involved, please cite the papers or other literature to amplify this point. I don't doubt this, but I do find that there needs to be some solid evidence to make this claim especially when we want to consider possibly using digital interventions in the pediatric population.

-"The success of interventions in studies using parents and caregivers may be explained by higher digital literacy and lower digital exclusion as compared to older populations..." I think another point of interest is to describe whether the adult studies included caregivers - comparing the effectiveness of digital interventions between caregivers and older adults is not a "fair comparison". Perhaps a discussion about whether there are studies that included caregivers for those older individuals because digital interventions are becoming a popular tool for involving all people in the circle of care.

3. Why do you think videos are the most common/ effective modality? Is it the visual aspects? The demonstration in the video, the pausing between segments etc? Think this thought can be further developed.

**Do you want your identity to be public for this peer review?** For information about this choice, including consent withdrawal, please see our Privacy Policy

Reviewer #1: No

**Figure resubmission:****Reproducibility:** To enhance the reproducibility of your results, we recommend that authors of applicable studies deposit laboratory protocols in protocols.io, where a protocol can be assigned its own identifier (DOI) such that it can be cited independently in the future. Additionally, PLOS ONE offers an option to publish peer-reviewed clinical study protocols. Read more information on sharing protocols at https://plos.org/protocols?utm_medium=editorial-email&utm_source=authorletters&utm_campaign=protocols

---

## [Editor Report · Decision Letter 2]

4 Feb 2025

The behavioural and cognitive impacts of digital educational interventions in the emergency department: a systematic review

PDIG-D-24-00184R2

Dear Dr. Turnbull,

We are pleased to inform you that your manuscript 'The behavioural and cognitive impacts of digital educational interventions in the emergency department: a systematic review' has been provisionally accepted for publication in PLOS Digital Health.

Best regards,

Jennifer N Avari Silva, MD

Section Editor

PLOS Digital Health